# Ethyl Vinyl Ketone Activates K^+^ Efflux to Regulate Stomatal Closure by MRP4-Dependent eATP Accumulation Working Upstream of H_2_O_2_ Burst in Arabidopsis

**DOI:** 10.3390/ijms23169002

**Published:** 2022-08-12

**Authors:** Junqing Gong, Lijuan Yao, Chunyang Jiao, Zhujuan Guo, Shuwen Li, Yixin Zuo, Yingbai Shen

**Affiliations:** National Engineering Research Center of Tree Breeding and Ecological Restoration, College of Biological Sciences and Technology, Beijing Forestry University, No. 35, Qinghua East Road, Beijing 100083, China

**Keywords:** ethyl vinyl ketone, eATP, H_2_O_2_, K^+^ efflux, DORN1-RBOHF, stomatal closure

## Abstract

Plants regulate stomatal mobility to limit water loss and improve pathogen resistance. Ethyl vinyl ketone (evk) is referred to as a reactive electrophilic substance (RES). In this paper, we found that evk can mediate stomatal closure and that evk-induced stomatal closure by increasing guard cell K^+^ efflux. To investigate the role of eATP, and H_2_O_2_ in evk-regulated K^+^ efflux, we used Arabidopsis wild-type (WT), mutant lines of *mrp4*, *mrp5*, *dorn1.3* and *rbohd/f*. Non-invasive micro-test technology (NMT) data showed that evk-induced K^+^ efflux was diminished in *mrp4*, *rbohd/f*, *and dorn1.3* mutant, which means eATP and H_2_O_2_ work upstream of evk-induced K^+^ efflux. According to the eATP content assay, evk stimulated eATP production mainly by MRP4. In *mrp4* and *mrp5* mutant groups and the ABC transporter inhibitor glibenclamide (Gli)-pretreated group, evk-regulated stomatal closure and eATP buildup were diminished, especially in the *mrp4* group. According to qRT-PCR and eATP concentration results, evk regulates both relative gene expressions of *MRP4/5* and eATP concentration in *rbohd/f* and WT group. According to the confocal data, evk-induced H_2_O_2_ production was lower in *mrp4*, *mrp5* mutants, which implied that eATP works upstream of H_2_O_2_. Moreover, NADPH-dependent H_2_O_2_ burst is regulated by DORN1. A yeast two-hybrid assay, firefly luciferase complementation imaging assay, bimolecular fluorescence complementation assay, and pulldown assay showed that the interaction between DORN1 and RBOHF can be realized, which means DORN1 may control H_2_O_2_ burst by regulating RBOHF through interaction. This study reveals that evk-induced stomatal closure requires MRP4-dependent eATP accumulation and subsequent H_2_O_2_ accumulation to regulate K^+^ efflux.

## 1. Introduction

Plant organisms, exposed to unusual and unpredictable environmental conditions, become more vulnerable to infections and pests as a result of non-optimal growing circumstances [1]. Stomata are gas-exchange holes on leaves that affect important functions such as photosynthesis and drought tolerance and are infections’ primary entry point [2]. 

Insect feeding, mechanical damage, and pathogen infection can cause the release of volatile organic compounds. Plant volatiles (VOCs) usually play important roles in regulating the feeding of insects and microorganisms on plants [3]. The esters in VOCs have been discovered to change the opening and closing condition of stomata in plants’ resistant response to harmful bacteria. Plants demonstrated stomatal closure after treatment with (*Z*)-3-hexylpropionate (HP) and (*Z*)-3-hexylbutyrate (HB) [4]. 

The volatiles containing α, β-unsaturated carbonyls are collectively referred to as reactive electrophilic substances (RES) [5,6]. (*E*)-2-hexenal, an electrophilic substance, synthesizes and releases rapidly during insect feeding [7] or bacterial infection [8,9]. Malondialdehyde (MDA), the smallest molecule contains α, β-unsaturated carbonyl group, increased exogenous jasmonic acid (JA), and salicylic acid (SA) in seedlings during Botrytis cinerea (*Botrytis cinerea*) infection [10]. In addition, OPDA, an electrophilic substance, is not only a precursor of JA but also can be used as a signal molecule to induce the expression of a large number of genes except JA-related genes [11,12,13]. Mattick and Hand initially detected evk in soybean homogenate (*Glycine max* (L.) *Merr.*) [14]. When LOX activity was blocked, soybeans were able to synthesize evk in a 13-HOPT-dependent way [15]. Evk contains α, β-unsaturated carbonyl structure of volatile small molecular substances that could activate defense gene expression in Arabidopsis [16]. When Arabidopsis was fed by *Pieris rapae*, evk, as pest-induced plant volatiles, would be induced to release [17]. Under oxygen stress, the freeze–thaw damaged soybean leaves will release a large amount of evk [18]. Increasing the content of evk by applying selenium at the green flower stage of tomato can reduce the incidence of flower end rot [19]. However, little is known about whether and how evk can cause stomata closure in Arabidopsis.

Adding ATP to leaf stomata caused fast closure and was implicated in biotic and abiotic stress responses [20,21,22,23,24]. ATP, known as an essential, cellular energy source to drive many biochemical reactions, which can be released into the extracellular, is referred to as extracellular ATP (eATP). eATP is synthesized inside the cell and secreted to the extracellular space through anion channels or by exocytosis [21,25,26,27], and the ABC transporters (ATP-binding cassette transporters) are probably involved in this transport process [22]. In Arabidopsis, MRP4, and MRP5, the multidrug resistance-associated proteins (MRPs) of ABC transporters, are located in guard cells and function in stomatal movements [28,29,30]. MRP5 shows partial inhibition of ABA-induced stomatal closure [28,29]. However, it is unclear whether evk can stimulate eATP accumulation and whether MPR participates in stomatal closure.

H_2_O_2_ promotes stomatal closure to avoid plant pathogen infection [31,32]. In Arabidopsis, ROS production in response to pathogen recognition (e.g., due to recognition of pathogen-associated molecular patterns (PAMPs)) is mediated by RBOHD with RBOHF showing some redundant functions [33]. NADPH oxidase is a homologue of RBOHs and plays an important role in ROS production network in plants. Studies have shown that RBOHF is a key regulator in response to ethylene stomata in guard cells [34]. Evidence suggests that melatonin-mediated and SOS-mediated Na^+^ efflux may be influenced by RBOHF-dependent ROS burst [35]. Studies have shown that RBOHD and RBOHF act together in guard cells in response to plant defense in the ROS-dependent ABA signal transduction pathway [33,36]. eATP and reactive oxygen species (ROS) crucially function in establishing plant developmental and adaptive plasticity [37,38,39,40]. However, it is unclear whether evk induces ROS or who is capable of affecting it, and whether H_2_O_2_ burst is involved in the process of stomatal closure caused by evk.

Outward rectifying potassium (K^+^ out) channels in guard cells are key factors for stomatal closure in leaves [41]. Potassium (K^+^) is one of the most abundant macronutrients in plants, accounting for up to 10% of the dry mass of the plant. Photosynthesis, plant development, and stress tolerance are all harmed by long-term K^+^ depletion [42,43,44]. During the stomatal opening, guard cells can collect up to a few hundred mmol of potassium [45,46]. Stomatal mobility is quantitatively connected to K^+^ transfer, according to early reports [47]. Stomatal mobility is primarily controlled by the swelling and shrinking of the surrounding guard cells, which is triggered by the net influx and efflux of solutes such as potassium (K^+^). K^+^ is a key osmolyte for cell turgor and membrane electric potential modulation [48,49]. 

Stomatal closure is mediated by eATP, H_2_O_2_, and K^+^, although it is unclear if evk may trigger stomatal closure or whether they are involved in the process. In the current study, evk causes stomatal movement by regulating guard cell K^+^ flux. Evk treatment enhanced the expression of ATP-related genes in Arabidopsis leaves. Using wild-type Arabidopsis and ABC transporter mutants, we investigated the involvement of eATP in evk-regulated stomatal movements. Using mutant *rbohd/f* we tested eATP accumulation and MRP4/5 relative gene expressions and confocal test in *mrp4*, and *mrp5* mutants, and evk-induced H_2_O_2_ production was diminished, indicating eATP is upstream of H_2_O_2_ in the process of evk-induced stomatal closure in Arabidopsis. The crosstalk between eATP, H_2_O_2_, and the K^+^ channel during stomatal movements was also invested.

## 2. Results

### 2.1. MRP4/5-Dependent eATP Buildup is Required for Evk-Induced Stomatal Closure

#### 2.1.1. Evk-Induced Stomatal Closure Was Suppressed by Gli, PPADS, and Apyrase in Arabidopsis

Stomata maintain moisture balance and protect plants from pathogen infection. Abaxial epidermal peels were peeled off from young fully expanded rosette leaves of 4-week-old Arabidopsis plants and pretreated under light for 2 h in stomatal opening solution, then evk was added to the test solution. The apertures were photographed and measured at 0 min, 30 min, and 60 min. The findings revealed that 8 μM evk caused stomata to close in WT plants (Figure 1a). To see if ABC transporter-dependent eATP generation is involved in evk-induced stomatal closure, we explored the impact of the ABC transporter inhibitor Gli, the eATP receptor DORN1 inhibitor PPADS, and the eATP hydrolytic enzyme Apyrase on evk-induced stomatal closure. Stomatal aperture was 0.21 ± 0.06 after 1 h treatment of evk which was greatly reduced compared to 0 min (0.59 ± 0.05), while in Gli/PPADS/Apyrase-treated group, stomatal aperture was 0.49 ± 0.02, 0.47 ± 0.02, 0.46 ± 0.03, respectively. Individual administration of one of these three reagents had a considerable effect on stomatal closure, and evk-induced stomatal closure was greatly reduced (Figure 1b). According to the findings, which suggested that MRP proteins, eATP, and DORN1 may be involved in the evk-induced stomatal closure signal process.

#### 2.1.2. eATP Accumulation Was Induced by Evk and Reversed by Gli

We assessed the change in eATP concentration in leaves treated with evk to confirm that eATP transported by ABC transporter is involved in evk-induced stomatal closure. The results revealed that evk considerably increased the eATP level in Arabidopsis seedling leaves with a peak of 523.93 ± 37.13 nM at 6 min. In the inhibitor Gli group, the eATP concentration reached a peak of 97.17 ± 7.62 nM at 6 min. The ABC transporter inhibitor Gli has the potential to reverse the evk-induced eATP increase (Figure 2a). These findings imply that evk may promote eATP and that ABC transporters may play a role in this process. 

#### 2.1.3. Evk-Induced eATP Accumulation and Stomatal Closure Were Impaired in Mrp4, Mrp5 Mutants

There was no significant difference in the size of stomatal openings between the *mrp4*, *mrp5* mutants and the WT Arabidopsis under control conditions. The stomata of WT plants closed after treatment with evk; however, this phenomenon was impaired in *mrp4* and *mrp5* mutants (Figure 1a,b). After soaking in the evk solution, the eATP concentration of WT keep rising at 1 min, 3 min, and 6 min and reached a peak of 523.93 ± 37.13 nM at 6 min, then slowly declined. When MRP4 is mutated, evk can hardly mediate eATP accumulation with a peak of 138.7 ± 5.4 nM, which is similar to the Gli-pretreated group (Figure 2a,b). When MRP5 is mutated, evk can mediate eATP accumulation with a peak of 334.32 ± 7.97 nM, which is lower than the WT + evk group (Figure 2b). When mutating MRP4, evk can hardly mediate stomatal closure, but in mutant *mpr5*, stomas can also slightly close. These findings revealed that MRP4 and MRP5 are engaged in the evk-induced stomatal closure process, especially MRP4 plays a more important role in evk-induced eATP buildup.

#### 2.1.4. Evk Treatment Enhanced the Expression of ATP-Related Genes in Arabidopsis Leaves

In order to test whether eATP is secreted via ABC transporters in evk-induced eATP accumulation, the effect of evk on the gene expression of *MRP4* and *MRP5* was examined. Primer sequences are as follows (Table 1). The qRT-PCR results showed that the gene expression of *MRP4* and *MRP5* was highly induced. Their expression peak was a 3.80-fold change, and 2.98-fold change, respectively, which occurred 10 min after evk treatment (Figure 3a). It suggested that evk induced the gene expression of *MRP4* and *MRP5* in Arabidopsis leaves, thereafter, more ATP was transported outside. Injury, pathogen infection, or osmotic stress can cause cells to release ATP, causing *Apyrase1* (*APY1*) and *Apyrase2* (*APY2*) gene expression to increase [50], showing that their expression is linked to eATP concentration regulation. The concentration of eATP is regulated in part by an enzyme called Apyrase (ecto-nucleoside triphosphate diphosphohydrolase). We confirmed Apyrase expression levels using qRT-PCR, the results showed that: *APY1* and *APY2* gene expression was significantly upregulated with a peak of a 3.22-fold change, and 2.54-fold change, respectively, after 30 min evk soaking (Figure 3b). Especially, *APY1* may play a major role in the degradation of eATP. DORN1 (DOESN’T RESPOND TO NUCLEOTIDES), extracellular binding domain binds to ATP, reached the highest expression value (3.93-fold change) at 30 min (Figure 3b). Plants respond to eATP with many of the same responses seen in animals, including the production of reactive oxygen species1 (ROS). The results showed that: *RBOHF* and *RBOHD* gene expression was upregulated after evk treatment, especially *RBOHF* (6.80-fold change) (Figure 3b). The upregulated expression of *MRP4/5*, *APY1/2*, *RBOHD/F,* and *DORN1* genes suggested that evk stimulated the participation of eATP and the activation of downstream H_2_O_2_. 

### 2.2. eATP is Up Stream of H_2_O_2_ in the Process of Evk-Induced Stomatal Closure in Arabidopsis

ATP can also cause rapid and dose-dependent increases in H_2_O_2_ generation in cultures. H_2_O_2_ is also involved in eATP-induced stomatal opening in Arabidopsis [33]. Is there any connection between eATP and H_2_O_2_ in Arabidopsis stomatal closure triggered by evk? This question was answered by further data.

#### 2.2.1. Evk-Induced H_2_O_2_ Burst is NADPH-Dependent

H_2_O_2_ functions as an important substance that mediates stomatal closure. We tested the variation in H_2_O_2_ levels using H_2_DCF-DA, a specialized H_2_O_2_ sensor, for detecting H_2_O_2_ levels in guard cells by performing confocal laser scanning microscopy from evk-treated versus untreated WT plants. H_2_O_2_ levels were strongly increased (9.04% above control) in guard cells from evk-treated WT plants compared to the control. In contrast, in evk-treated guard cells pretreated with DPI (diphenyleneiodonium chloride) and in cells of the NADPH oxidase mutant *rbohd*, *rbohf,* and *rbohd/f*, H_2_O_2_ enhanced levels (respectively, about 2.39%, 2.46%, 1.69% above control) were significantly lower than WT treated with evk group (Figure 4a). These results indicate that RBOHD/F functions in evk signal transduction. 

#### 2.2.2. The Effects of Gli, PPADS, and Apyrase on Evk-Induced H_2_O_2_ Synthesis in Arabidopsis 

The effects of Gli, PPADS, and Apyrase on evk-induced H_2_O_2_ synthesis in Arabidopsis leaf guard cells were investigated in terms of understanding the relationship between eATP and H_2_O_2_ in the process of evk-induced stomatal closure. When WT guard cells were treated with evk, H_2_O_2_ fluorescence rapidly increased (Figure 4a). However, in PPADS-treated WT stomas, Gli-treated WT stomas, and Apyrase-treated WT stomas, only slight increases (about 2.5–3.5% above control) in H_2_O_2_ concentrations were detected in response to evk treatment (Figure 4a,c). The results showed that Gli, PPADS, and Apyrase prevented the activation of evk on H_2_O_2_ synthesis in Arabidopsis guard cells (Figure 4a,c), indicating that eATP may act upstream of H_2_O_2_ in evk-induced stomatal closure. 

#### 2.2.3. In Mrp4, Mrp5 Mutants, Evk-Induced H_2_O_2_ Production Was Diminished

H_2_O_2_ was identified in the leaves of *mrp4* and *mrp5* after treatment with 8 μmol/L evk for 30 min. The results showed that evk significantly increased H_2_O_2_ (8.36% above control) (Figure 4b) in WT guard cells. We discovered that only slight increases in the guard cells of *mrp4* and *mrp5* mutants (about 3.7–4.3% above control) in H_2_O_2_ concentrations were detected in response to evk treatment (Figure 4b,d), indicating that MRP4 and MRP5 are involved in evk-induced H_2_O_2_ production.

#### 2.2.4. Mutations in RBOHD/F did not Affect Evk-Mediated eATP Accumulation and Evk-Mediated MRP Upregulation

Our findings show that eATP mediates evk-induced stomatal closure by acting upstream of H_2_O_2_. Does H_2_O_2_ affect the secretion of eATP? The involvement of H_2_O_2_ in evk-induced eATP release and the effects of evk on relative gene expression of *MRP4* and *MRP5* in *rbohd/f* double mutant leaves were investigated in line with the research questions. The eATP concentration in seedling leaves of WT increased with a peak of 523.93 ± 37.13 nM. Moreover, the eATP concentration in seedling leaves of *rbohd/f* on evk-induced eATP secretion increased 489.87 ± 31.52 nM, with no big difference discovered (Figure 2a). With evk treatment, the relative gene expression of *MRP4* and *MRP5* in the leaves of WT was upregulated with peaks of 3.80-fold change and 2.98-fold change, respectively. The effect of evk on *MRP4* and *MRP5* transcription in the leaves of the *rbohd/f* double mutant were also investigated. The qRT-PCR results (Figure 2b) showed that evk still regulates the relative gene expression of *MRP4* and *MRP5* in the leaves of the double mutant *rbohd/f* with peaks of 4.12-fold change and 2.43-fold change, respectively, implying that eATP functions upstream of H_2_O_2_ to participate in the evk-induced stomatal closure.

### 2.3. In Wild Arabidopsis Guard Cells, Evk Stimulated Outward K^+^ Currents, but This Activation Was Impaired in Mrp4 and Rbohd/f Mutants

To induce stomatal closure, does evk control K^+^ current by influencing the K^+^ channels in guard cells? The K^+^ currents in wild type, *mrp4* mutant, and *rbohd/f* mutant guard cells were measured using the NMT technique. Based on the transient processing of K^+^ ions by evk, the test time was separated into three sections: pre-evk treatment (pre), evk-responsive peak (peak), and post-evk response (post). The results revealed that the amplitude of the basal current did not differ significantly between these lines. After evk treatment, K^+^ flow in WT guard cells peaked at 829.54 ± 59.9 pmol cm^–2^ s^–1^ and subsequently gradually decreased, by 3 min, the K^+^ flow had stabilized and had moved into the post-period (Figure 5a). Treatment with evk promoted the outward-rectifying K^+^ currents of the plasma membrane in wild guard cells, but in the guard cells of *mrp4* (peaked at 264.59 ± 18.65 pmol cm^–2^ s^–1^), *rbohd/f* (peaked at 397.88 ± 35.91 pmol cm^–2^ s^–1^) and *dorn1.3* mutants (peaked at 166.71 ± 23.76 pmol cm^–2^ s^–1^), K^+^ efflux decreased significantly (Figure 5a). As shown in Figure 5b, evk caused significant net K^+^ efflux in WT guard cells during the peak response period. Then, the net K^+^ flux recovered to the levels before exposure to evk. In *mrp4*, *rbohd/f*, and *dorn1.3* guard cells, the evk-induced net K^+^ efflux in the peak response period was significantly lower than that in WT guard cells. The findings suggested that evk regulates stomatal movements by modulating the net content of solute and osmolarity in the cytosol by influencing the activation of K^+^ channels in an eATP- and H_2_O_2_-dependent way.

### 2.4. RBOHF Interacts with DORN1

According to the confocal data, in the mutant *rbohf* and *dorn1.3*, the fluorescence of H_2_O_2_ increased by 2.45% and 2.27%, respectively, after evk was applied, the H_2_O_2_ burst caused by evk is participated by a kinase DORN1 and RBOHF. It is possible to regulate RBOHF through DORN1, so as to regulate the ROS burst. The interaction between the two proteins is illustrated by yeast two-hybrid (Y2H), firefly luciferase complementation imaging assay (LCI), pulldown, and BiFC experiments. In a Y2H experiment, DORN-BK has self-activation, which can be effectively inhibited after inhibition by 12.5 mM 3-AT, and there is interaction between DORN-BK and RBOHF-N-AD (Figure 6a). In vivo experiments showed that DORN-YNE and RBOHF-YCE could interact on the cell membrane of tobacco (Figure 6b). The results in vitro are the same. In the anti-His pulldown line, RBOHF-His was not detected in RBOHF-His with GST group, but in RBOHF-His with DORN-GST group, RBOHF-His was detected by immunoblot analysis using anti-HIS antibodies (Figure 6c). For further confirmation, DORN1-Cluc and RBOHF-Nluc were injected into tobacco leaves at the same time by Agrobacterium. We found that on leaves which were divided into four parts (Cluc/Nluc, DORN1-Cluc/Nluc, Cluc/RBOHF-Nluc, DORN1-Cluc/RBOHF-Nluc), the DORN1-Cluc/RBOHF-Nluc parts had stronger fluorescence than other three control parts (Figure 6d), indicating that the interaction can be realized in vivo and in vitro. It suggests that DORN1 may control the burst of ROS by regulating the activity of RBOHF through interaction.

### 2.5. Evk Reduced Stomatal Conductance

After fumigation with evk, there was no significant difference in the dry/fresh ratio between evk and the control group, indicating that evk did not cause stress on plant growth (Figure 7a). Compared with the control group, the stomatal conductance of the evk treatment group decreased significantly, and the stomatal conductance of the mutant also decreased, but the degree of decline was less than that of the wild type (Figure 7b). Measurements of stomatal conductance showed similar results to the stomatal aperture experiments (Figure 1a). It indicates that MRP4, RBOHD/F, and DORN1 are involved in the process of evk-mediated stomatal closure. 

## 3. Discussion

α, β-unsaturated carbonyl is an important structure of long-distance signal transduction in plants. The long-distance signal transduction mediated by GLVs may also contain α, β-unsaturated carbonyl [6]. Evk as a plant volatile, as well as RES, would be induced to release under stresses such as Pieris rapae feeding in Arabidopsis, the freeze–thaw damaged soybean leaves [17,18]. Evk can reduce the incidence of flower end rot [19]. However, little is known about whether and how evk can cause stomata closure in Arabidopsis. Evk contains α, β-unsaturated carbonyl structure, has the ability to move vast distances, and controls stomatal closure (Figure 1a).

Firstly, K^+^ efflux will lead to stomatal closure. The study found that evk can mediate K^+^ efflux, thereby mediating stomatal closure. So, whether eATP and H_2_O_2_ are involved in the stomatal closure caused by evk? We first found through biological experiments that inhibition of MRP protein to accumulate eATP (Gli treatment), inhibition of eATP receptor DORN1 (PPADS treatment), degradation of eATP (Apyrase treatment), inhibition of NADPHase to produce H_2_O_2_ (DPI treatment) could weaken evk-mediated stomatal closure (Figure 1a,b), and evk mediated upregulation of ATP related genes and H_2_O_2_ related genes (Figure 3a,b), which indicates that ATP and H_2_O_2_ are involved in evk mediated stomatal closure. 

Evk can mediate the accumulation of eATP through MRP4 and trigger *MRP4* and *MRP5* relative gene expression indicating that evk can activate rather than inhibit MRP4 (Figure 3a). Curcumin, which contains two electrophilic compounds α, β-unsaturated carbonyl group, inhibits MRP by curcumin and MRP protein interaction [51]. It has also been confirmed that curcumin significantly inhibits MRP1 and MRP2 mediated transport in the isolated membrane capsule of Sf9 cells expressing MRP1 and MRP2. Curcumin also inhibits MRP1 mediated activity in the intact MRP1 overexpressed Madin Darby canine kidney (mdckii-mrp1) cell monolayer. Trans-2-hexanal, which contains one electrophilic compound α, β-unsaturated carbonyl group did not inhibit MRP1 [51]. Drugs containing this group may not all inhibit or activate MRP proteins. Whether evk really activates MRP4 needs more professional experiments.

Secondly, eATP and H_2_O_2_ are involved in evk-mediated stomatal closure. What is their upstream and downstream relationship? Through qRT-PCR experiment and confocal experiment, it was found that when RBOHD/F was mutated, it did not affect evk mediated eATP accumulation or MRP protein expression (Figure 2a,b). However, the treatment of Gli, PPADS, and Apyrase prevented the activation of evk on H_2_O_2_ synthesis in Arabidopsis guard cells (Figure 4a–d). In summary, eATP is upstream of H_2_O_2_ in the process of evk-induced stomatal closure in Arabidopsis. This result is consistent with previous studies [52].

Mutation of DORN1 phosphorylation sites on RBOHD eliminates the ability of ATP to induce stomatal closure [20]. Evk also upregulated DORN1, RBOHF gene (Figure 3b), according to confocal data, mutation of DORN1, RBOHF showed a decrease in H2O2 burst compared to WT treated with evk group (Figure 4a,c), which indicated that DORN1 and RBOHF may be involved in evk-induced H2O2 burst. Moreover, the interaction between these two is illustrated by yeast two-hybrid (Y2H), firefly luciferase complementation imaging assay (LCI), pulldown, and BiFC experiments (Figure 6a–d), indicating they can be realized in vivo and in vitro. The expression of heterologous RBOHD in HEK293T cells shows that direct phosphorylation or Ca^2+^ binding to EF chiral domain can synergistically activate RBOHD and produce ROS [34]. In addition to the regulation of Ca^2+^ and phosphorylation, more mechanisms have been found, such as 14-3-3, ROP regulation and sulfhydryl nitrite modification [53,54,55,56]. Some studies have shown that RBOHF can be directly regulated by Ca^2+^ or by CBL1/9–CIPK26 calcium dependent phosphorylation complex [57]. Both CIPK11 and CIPK26 can phosphorylate RBOHF-N in vitro and activate RBOHF in HEK293T cells [58]. SRC2 can activate the production of ROS mediated by Ca^2+^-dependent RBOHF in Arabidopsis [59]. OST1 is located upstream of ROS and can phosphorylate Ser13 and Ser174 sites of RBOHF [60]. In this study, we demonstrated that DORN1 and RBOHF can be realized in vivo and in vitro. RBOHF can be regulated by DORN1 through interaction.

Finally, are H_2_O_2_ and ATP involved in evk mediated K ion efflux? The rapid accumulation and release of K^+^ and of organic and inorganic anions by guard cells control the opening and closing of stomata and thereby gas exchange and transpiration of plants. A recent study identified an insertional T-DNA disruption mutant in the Arabidopsis guard cell K^+^ out channel gene, *GORK* [41], they provide direct genetic evidence that outward rectifying K^+^ out channels in guard cells contribute to stomatal closing in leaves. We also conducted NMT experiments. Evk promoted the outward-rectifying K^+^ currents of the plasma membrane in wild guard cells, but in the guard cells of *mrp4*, *rbohd/f*, and *dorn1.3* mutants K^+^ efflux decreased significantly, which implies that evk activation of outward K^+^ channels in an eATP- and H_2_O_2_-dependent way (Figure 5a,b). The findings suggested that evk regulates stomatal movements by modulating the net content of solute and osmolarity in the cytosol by influencing the activation of K^+^ channels in an eATP- and H_2_O_2_-dependent way.

The activity of GORK channels may be modulated by cyclic nucleotides, gamma-aminobutyric acid, G-proteins, protein phosphatases, inositol, ATP, reactive oxygen species, and nitric oxide [61,62,63,64]. We guess GORK can be activated by ATP, DORN1 kinase, or H_2_O_2_ during evk-mediated K^+^ efflux. More research into how to be activated is recommended. Because GORK is regulated by both ATP and H_2_O_2_, we believe GORK is involved in the evk-mediated K^+^ efflux pathway. This hypothesis will require more research to be confirmed.

In terms of experimental methods, three improved experimental methods are needed in future research. Firstly, a previous study demonstrated that DPI at subpicomolar concentrations (10^−13^ to 10^−14^ M) exhibits specific effects against NOX2 in primary midbrain neuron-glia cultures [65]. However, what is the dosage of DPI in plants? Through consulting the literature, we found that the concentration of NADPHase inhibition in plants is 100 µM [66], 10 µM [67], and 50 µM [68]. After WT was pretreated with DPI at different concentrations, it was treated with 8 µM evk for 1 h to see the stomatal aperture. It was found that the effect of evk on stomatal closure could be inhibited at the DPI concentration of 50 µM. Will the concentration in plants be too high? Will it lead to a toxic effect rather than a real inhibitory effect? This part needs further verification. Secondly, for the determination of ROS, it is necessary to complete the H_2_DCF-DA concentration-specific reaction curve, as well as the dyeing process and the de-esterification process.

Thirdly, determine the effect of evk on plant eATP concentration. Planting plants in 96-well plates will have an impact on plant growth, which is not conducive to plant growth. The nutrients obtained by the roots are very limited. Due to the space problem, only 180 uL of culture medium is added, so it will be subject to nutrient stress, and the overall growth of plants is relatively small. The advantage of this method is that it can detect the release of eATP concentration of seedlings by detecting the eATP in the living state, but its limitations cannot be ignored, and it needs to be improved here. However, the problem of the growth restriction of plants from the narrow space of the 96-well plates cannot be ignored, and I need to propose better research methods.

In conclusion, evk mediates the buildup of eATP mainly by MRP4 (Figure 8(1)). DORN1, as an eATP receptor, receiving the eATP signal interacts with downstream RBOHF to regulate the burst of H_2_O_2_ (Figure 8(2,3)) and then regulates the external emission of K^+^ (Figure 8(4)), which eventually leads to stomatal closure (Figure 8(5)). Finally, evk-mediated stomatal closure is accomplished by controlling K^+^ efflux via MRP4-dependent eATP release and subsequent NADPH-dependent ROS burst.

## 4. Materials and Methods

### 4.1. Plant Materials and Culture Conditions

Seeds of wild-type Arabidopsis (Col-0), and mutants *rbohd* (CS9555, AT5G47910), *rbohf* (CS9558, AT1G64060), *rbohd/f*, *dorn1.3* (CS859164, AT5G60300), *mrp4* (SALK_116959C, AT2G47800), *mrp5* (SALK_046377C, AT1G04120) were vernalized at 4 °C for 2 days in the dark. The seeds were surface-sterilized in 75 percent ethyl alcohol for 4 min after vernalization, then washed four times in sterile water, sowed in autoclaved soil mixture, and placed in an incubator (Percival model: I-36vl). Plants were cultivated in soil at 21–23 °C, at 70% relative humidity, and 80–110 mol m^–2^ s^–1^ light intensity under long-day (16 h light/8 h dark) conditions [69].

Detection of qRT-PCR, stomatal aperture measurements, K^+^ efflux was performed on 18-day-old Arabidopsis WT or mutant plants, which were planted in soil. Detection of eATP content and reactive oxygen species (ROS) in guard cells was analyzed on 12-day-old Arabidopsis WT or mutant plants. Seeds were surface sterilized and sown on Petri plates with 1/2 Murashige and Skoog (MS) solid medium (2.2 g L^−1^ MS, 10 g L^−1^ sucrose, 0.5 g L^−1^ MES, and 1.2% agar, pH 5.8 adjusted using KOH). The concentration of evk used in all experiments was 8 µM.

Nicotiana culture conditions: take the dry stored tobacco seeds into the centrifuge tube, add an appropriate amount of deionized water to clean the surface 3 times, and add 1ml of deionized water to soak in the tube for 24 h to make the seeds absorb water and prepare for germination. Prepare nutritious soil and vermiculite, mix them with water in the proportion of 3:1, and then put them into a plastic flowerpot with a diameter of 7 cm and a height of 8 cm. Absorb the soaked tobacco seeds with a rubber-tipped dropper and directly sow them in the flowerpot. The surface is covered with fresh-keeping film to reduce water evaporation during germination. Put it in the light incubator and cultivate it according to the temperature of 25 ℃, humidity of 60%, light intensity of 100 µ mol m^−2^ s^−1^, light cycle of 16 h light/8 h darkness. After a week, the seeds grow two leaves, remove the plastic wrap and continue to culture for a week. Select the seedlings with the same growth trend and transfer them to the plastic flowerpot with nutrient soil: vermiculite of 3:1 and continue to culture in the incubator under the same conditions for 3-week-old *Nicotiana benthamiana* used for firefly luciferase complementation imaging assay.

### 4.2. Measurement of Stomatal Aperture

In order to ensure that before the measurement of stomatal aperture, the stomata of plants were unified and completely open, tear off the lower epidermis and lay it on the bottom of the dish. Fix the lower epidermis of 4–5-week-old leaves blade with adhesive tape on a 35 mm dish incubated in 5 mL stomatal opening solution (30 mM KCl, 1 mM CaCl_2_, 10 mM Tris, pH 5.8) to expose to white light for at least 2 h, room temperature. After this incubation, evk was added to the buffer. The stomatal aperture was measured at 0 min, 60 min, and 90 min. The longitudinal length and lateral length were measured using Image-Pro Plus (Media Cybernetics Co., Silver Spring, MD, USA). At least 60 stomas were measured per epidermal peel. The ratio of transverse length and longitudinal length was calculated and used as an index of stomatal aperture.

### 4.3. Detection of eATP Content in Arabidopsis 

All seedings planted in the 96-well plate (one seedling per well) germinate in 3 days and then grow for 7 days. Seedlings that grow for 10 days are actually only 7 days old. The seedlings are in a relatively small state, so the whole plant can be covered with 100uL of test solution. For comparison, seedlings were simply submerged in 100 µL of test solution (0.1 mM KCl, 0.1 mM CaCl_2_, 0.1 mM MgCl_2_, 0.5 mM NaCl, 0.3 mM MES, 0.2 mM Na_2_SO_4_, pH 6.0) at the onset of the treatment. Wild-type seedlings were incubated in 100 µL of test solution, evk solutions, or solutions with Gli, respectively, and the mutant seedings were incubated in the test solution or evk solutions, and samples were immersed in different solutions for 1 min, 3 min, 6 min, 9 min, and 12 min. The final concentration of Gli and evk is 100 µM, 8 µM. For analysis of the eATP concentration, 50 µL of sample was mixed with 50 µL of Luciferase/Luciferin reagent (ENLITEN^®^ ATP Assay System, Promega, Madison, WI, USA), and the light emission at 420 nm was measured in a GloMaxTM 96 Luminometer, and subsequent luciferin–luciferase treatment and luminescence measurements were performed at the luminometer using a 5 s integration time [21]. The amount of eATP was calculated from a standard curve. Experiments were performed in triplicate.

### 4.4. qRT-PCR

Total RNA was isolated from leaves of Arabidopsis using an RNA extraction kit, and cDNA was produced using a reverse transcriptase kit for quantitative RT-PCR (Takara). *ACTIN2* and *EF1α* were used as references [69,70,71]. RT-PCR was performed in a 7500 fast real-time PCR system (Applied Biosystems, Foster City, California, USA) using a Power SYBR Green PCR Master Mix kit (Applied Biosystems, Foster City, CA, USA), and the 2^-ΔΔCt^ method was used to calculate relative gene expression levels. The expression levels of several important eATP-related genes were detected, including in Table 1.

### 4.5. H_2_O_2_ Detection in Guard Cells

Epidermal peels isolated from 4–5-week-old plants were fixed on the bottom of a Petri dish and then incubated in 30 mM KCl and 10 mM MES (pH 6.5) for 2 h in light (150 mmol m^–2^ s^–1^). To observe transient evk responses, the leaves of two-month-old seedlings (WT, WT pretreatment with DPI, *mrp4*, *mrp5*, *dorn1.3*, *rbohf*, *rbohd*, *rbohd/f*) were immerged in 50 µM H_2_DCF-DA for 12 min at 25 °C in the dark. Then, the leaves were steeped in liquid 1/2 MS and treated with 8 µM evk. After 30 min of evk processing, the sample will be placed on the microscope and observed samples under a microscope with excitation at 490 nm and emission at 530 nm. Dye-loaded guard cells were examined using a laser scanning confocal microscope (Leica TCS SP8, Leica Microsystems, Wetzlar, Germany). Each datum was obtained from at least 60 guard cells. The amount of H_2_DCF-DA fluorescence in guard cells was measured with Image-J.

The concentrations of DPI, PPADS, Apyrase, and Gli in the experiments were 50 µM, 100 µM, 25 µM, and 1.5 µM, respectively.

### 4.6. Evk Fumigation

Arabidopsis plants were fumigated in glass bell jars with height of 12.5 cm and diameter of 15 cm with evk (≥97 percent, obtained from Sigma-Aldrich, Shanghai, China, CAS NO: 1629-58-9). Cotton balls with a diameter of 1 cm were soaked in evk dissolved in ethyl alcohol (evk at a final concentration of 8 µM) and hung in the bell jars; control cotton balls were soaked in the same volume of ethanol. The bell jars were immediately sealed with Vaseline petroleum jelly after putting the cotton balls [66].

### 4.7. K^+^ Flux Measurements in Guard Cells

Non-invasive micro-test technology (NMT) was used to detect K^+^ flow in guard cells (BIO-001A, Younger LLC. Amherst, MA, USA). In the test, epidermal peels isolated from 18-day-old plants were fixed on the bottom of a 35 mm dish incubated in the test buffer (0.1 mM KCl, 0.1 mM CaCl_2_, 0.1 mM MgCl_2_, 0.5 mM NaCl, 0.3 mM MES, 0.2 mM Na_2_SO_4_, pH 6.0), for 20 min in light. Silanized glass micropipettes (2–4 m aperture) were filled with electrolyte solution (100 mM KCl) and then front-filled to about 20 µm with a selective liquid ion exchange (LIX) cocktail. For electrode calibrating, use a 0.05 mM, 0.1 mM, or 0.5 mM KCl solution. The glass micropipettes were connected to the NMT system with a silver chloride wire, and electrodes with Nernstian slopes of 58 ± 5 mV log^–1^ were used.

Then, the K^+^ flux of individual guard cells on the epidermal peels were measured using NMT [72]. The sensor was located above the guard cells (1–2 µm) and was controlled by stepper motors that oscillated along the *Z* axis above the guard cells at a distance of 10 µm, implying that the sensor oscillated between 1–12 mm above the guard cells. After about 5 min of measurement of the basic ion flux, evk was quickly added to the test solution to a final concentration of 8 µM, data were collected for approximately 2.5 min as the evk response peak (peak) group. Data were then collected from the post-evk response (post) group to indicate the end of the reaction. The final flux values are reported as the mean of eight individual plants per treatment. The K^+^ flux was calculated using Fick’s law:$$J=-D\frac{\Delta C}{\Delta X}$$

*J* is the flux of K^+^ (pmol cm^–2^ s^–1^);D is the diffusion coefficient (1.96 × 10^–5^ cm^2^ s^–1^);Δ*C* is the difference between the concentrations near and far from the cells;Δ*X* is 10 µm. Each group contained 3–4 replicates.

### 4.8. Yeast Two-Hybrid (Y2H) Assay

The sequences encoding amino acid residues 1–387aa of RBOHF (RBOHF(1–387aa)) were cloned into pGADT7. The sequences encoding DORN1 were cloned into pGBKT7. The plasmid pairs RBOHF(1–387aa)-AD plus DORN1-BK were co-transformed into competent yeast strain AH109 cells. The cells were grown on SD/–Leu/–Trp medium. After 4 days, growing colonies were transferred to SD/–Ade/–His/–Leu/–Trp for further verification. Growing colonies on SD/–Ade/–His/–Leu/–Trp were transferred to SD/–Ade/–His/–Leu/–Trp with X-*α*-gal. Cell growth on SD/–Leu/–Trp and SD/–Ade/–His/–Leu/–Trp and blue colonies on SD/–Ade/–His/–Leu/–Trp with X-α-gal indicate protein–protein interactions between the two proteins. 12.5 mM 3-AT could be used to inhibite self-activation.

### 4.9. Firefly Luciferase Complementation Imaging (LCI) Assay 

Coding sequences of DORN1 were cloned into the Cluc plasmid. Coding sequences of RBOHF were cloned into the Nluc plasmid, and the constructed plasmids were transformed into Agrobacterium strain GV3101. A single colony was put into the resistant YEB liquid medium (beef extract 10 g, yeast extract 10 g, NaCl 5 g, pH 7.0) and cultivated overnight at 28 °C, 200 rpm. After 2 min of centrifugation at 12,000 rpm, the medium was discarded. The bacterium was washed five times in tobacco transformation solution (10 mM MES, 10 mM MgCl_2_, pH 5.6), resuspended in tobacco transformation buffer containing 0.1 mM acetosyringone (AS), and infiltrated into the leaves of 3-week-old Nicotiana benthamiana leaves. Fluorescence from luciferase in Nicotiana tabacum leaves infected with Agrobacterium was imaged with a molecular imaging system (LB983, Berthold Technologies, Bad Wildbad, Germany). Each leaf was divided into four quadrants prior to injection with the following combinations: Cluc/Nluc, DORN1-Cluc/Nluc, Cluc/RBOHF-Nluc, and DORN1-Cluc/RBOHF-Nluc. The experiments were repeated at least three times.

### 4.10. In Vitro Pulldown Assay

The genes encoding 344–616aa DORN1 were cloned into pGEX-4T-1 vector via EcoRI/XhoI, and the genes encoding 1–387aa RBOHF were cloned into pET28A vector via EcoRI/XhoI. DORN1-GST and RBOHF-HIS were transformed into E. coli Rosetta (DE3) cells for protein expression. DORN1-GST was used as the bait protein. Protein-His was as the capture protein. Glutathione beads containing 50 µg DORN1-GST or GST were incubated with 50 µg RBOHF-His in pulldown buffer (1% NP40, 150 mM NaCl, 50 mM Tris–HCl, 1 mM EDTA, pH 7.5, Add protease inhibitor before use) at 4 °C for 2 h, respectively. The protein beads complexes were washed with pulldown buffer, then centrifuged 500× *g* for 5 min, 5 times; then, add 150 µL elution buffer (250 µL 1 M Tris-HCl, 30 mg Glutathione, pH 8.0). Elute at room temperature for 30 min. Centrifuge 500× *g* for 5 min, suck the supernatant and add 5× boiling in loading buffer for 5 min and centrifuging at 12,000 rpm for 5 min, at last, the SDS sample buffer was boiled for 10 min. The binding of DORN1-GST with RBOHF-His was detected by immunoblot analysis using anti-GST and anti-His antibodies.

### 4.11. Bimolecular Fluorescence Complementation Assay

DORN1 was cloned into the pSPYNE vector via BamHI/XhoI, RBOHF was cloned into the pSPYCE vector via BamHI/XhoI, and the plasmids were transformed into Agrobacterium strain GV3101. Both constructs (DORN1-YNE and RBOHF-YCE) were transformed to Nicotiana tabacum leaves infected with Agrobacterium. YFP fluorescence was imaged with confocal laser scanning microscopy (Leica TCS SP8, Leica Microsystems, Wetzlar, Germany) at an excitation wavelength of 510 nm and an emission wavelength of 510–530 nm. 

### 4.12. Fresh/Dry Weight Measurements

Arabidopsis plants treated with ethyl alcohol and evk (evk’s final concentration was 8 μM) were used to measure fresh weight. Additionally, the samples were then dried in an oven at 80 °C for 24 h to constant weight and weighed as dry weight. Every group contained 6–8 Arabidopsis plants.

### 4.13. Measurement of Stomatal Conductance

Leaf stomatal conductance was measured with a diffusion porometer (Model AP4, Delta-T Devices, Cambridge, UK), taking measurements from at least six leaves per treatment according to the methods of Nogues and Baker (1998) [69].

### 4.14. Statistical Analysis

Dunnett’s C (variance not neat) at the level of *p* < 0.05 was significant. Error bars denote ± standard error of mean (SEM).

Accession numbers: The GenBank numbers and Genome Initiative numbers of all genes used in this article are as follows: RBOHD (GenBank NM_124165, AT5G47910), RBOHF (GenBank NM_105079, AT1G64060), DORN1 (GenBank NM_125423, AT5G60300), MRP4 (GenBank NM_130347, AT2G47800), MRP5 (GenBank NM_100293, AT1G04120), APY1 (NM_111279, AT3G04080) APY2 (NM_121833, AT5G18280), ACTIN2 (GenBank NM_001338358, AT3G18780), EF1α (GenBank NM_001125992.1, AT5G60390).

## Figures and Tables

**Figure 1 ijms-23-09002-f001:**
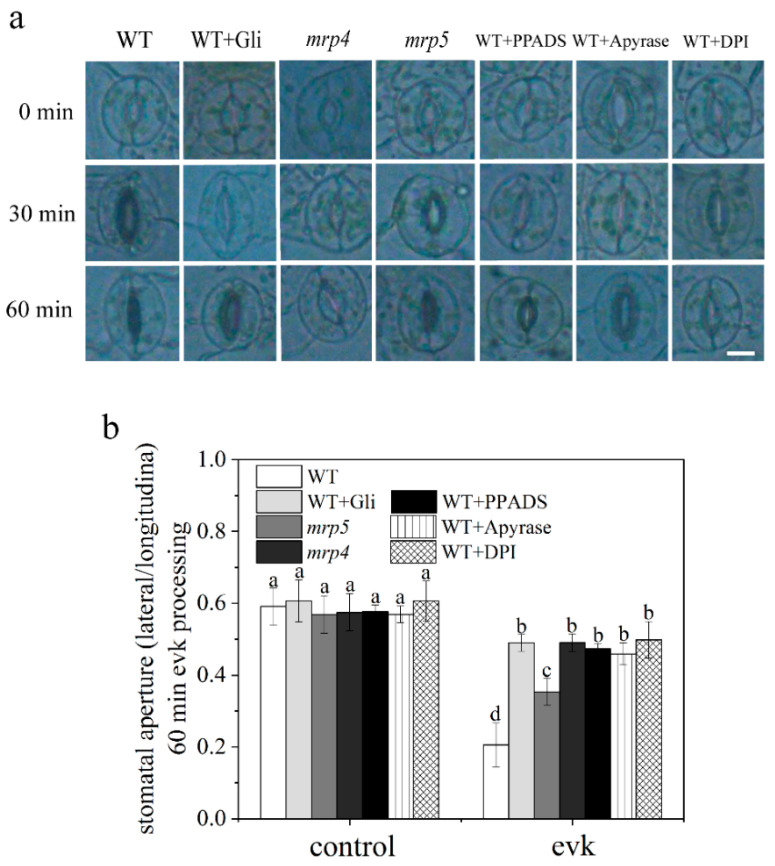
Time–response curves of evk induced stomatal closing in Arabidopsis. (**a**) Representative stomatal aperture of guard cells at 0 min, 30 min, and 60 min. Scale bar = 10 μm. Gli (the inhibiter of ABC transporter), PPADS (the inhibiter of eATP receptor DORN1), DPI (the inhibiter of NADPHase), and the Apyrase (eATP degradation) partially impaired the function of evk-induced stomatal closure. The point before evk treatment was treated as 0 min. (**b**) Histogram of stomatal aperture after 60 min evk treatment. The vertical scale represents the ratio of lateral length and longitudinal length. Data were obtained from at least 60 guard cells from different plants. Bars represent the standard error of the mean.

**Figure 2 ijms-23-09002-f002:**
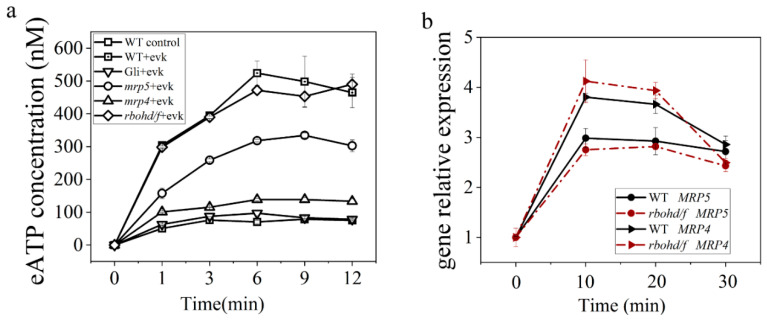
Evk-induced eATP accumulation requires MRP4 and MRP5, and H_2_O_2_ may work downstream of eATP signal. (**a**) The concentrations of eATP at different time points under evk soaking were measured in WT, Gli-treated WT group, mutants *mrp4*, *mrp5* and *rbohd/f* in 1 min, 3 min, 6 min, 9 min, 12 min. Evk increased secreting of eATP in seedling leaves of Arabidopsis while Gli reversed the evk-induced eATP elevation. MRP4 and MRP5 involed in evk-induced eATP accumulation. The accumulation of eATP induced by evk was not weakened in the mutant *rbohd/f*. Each group had six replicates. (**b**) Evk-enhanced the expressions of *MRP4* and *MRP5* were not weakened in the mutant *rbohd/f*. Each group had three replicates, and bars represented standard error of the mean. Error bars denote ± SEM, *n* ≥ 3.

**Figure 3 ijms-23-09002-f003:**
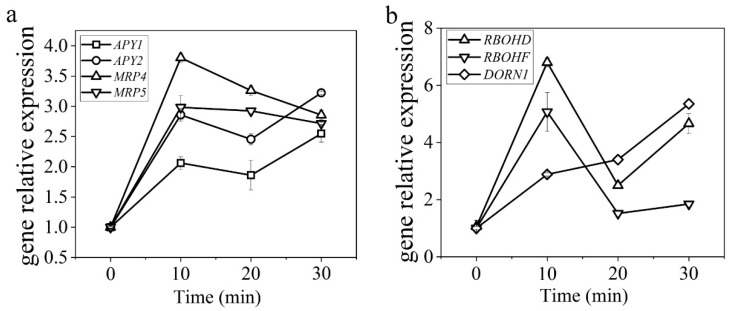
Expression analysis of eATP/H_2_O_2_-related genes. (**a**,**b**) Relative gene expression levels of *MRP4*, *MRP5*, *APY1*, *APY2*, *DORN1*, *RBOHD*, *RBOHF* in leaves of two-week-old WT Arabidopsis under evk treatment and each group had three technical replicates, and bars represented standard error of the mean. Error bars denote ± SEM, *n* ≥ 3.

**Figure 4 ijms-23-09002-f004:**
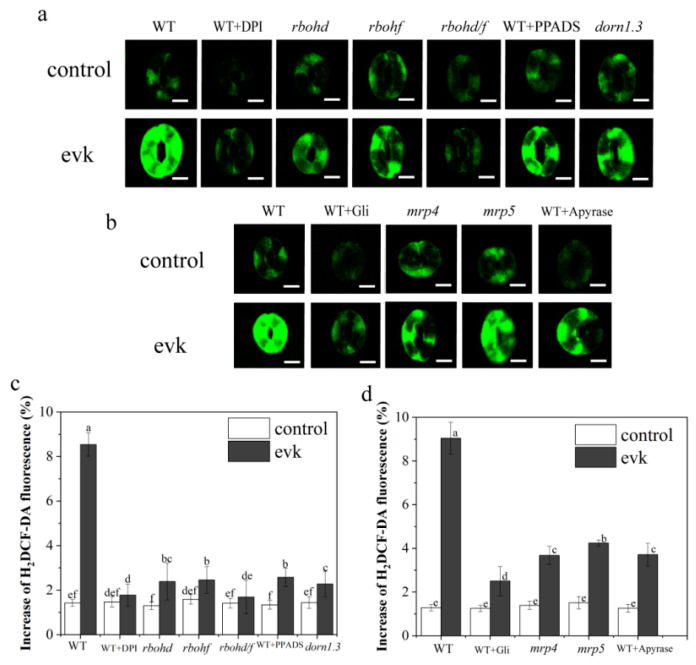
Evk-induced H_2_O_2_ generation in guard cells is NADPH-dependent and eATP may work upstream of H_2_O_2_ burst. (**a**) Evk-induced H_2_O_2_ generation in WT, WT pre-treated with DPI, mutant *dorn1.3*, *rbohd*, *rbohf*, and *rbohd/f* guard cells. The vertical scale represents the relative increase in H_2_DCF-dependent fluorescence. The H_2_DCF-dependent fluorescence of guard cells before 8 μM evk treatment was normalized to the control value. Data were obtained from at least 60 guard cells and bars represent standard error of the mean. (**b**) Photographs were taken from representative epidermal strips loaded with H_2_DCF-DA in Gli-pretreated WT, Apyrase-pretreated WT, WT, *mrp4*, and *mrp5* lines, respectively. (**c**) Evk-induced increases in H_2_O_2_ fluorescence were significantly suppressed in DPI-treated WT, mutant *rbohd*, *rbohf*, *rbohd/f*, *dorn1.3*, and PPADS-treated WT cells compared to evk-treated WT control cells. (**d**) Evk-induced increases in H_2_O_2_ fluorescence were significantly suppressed in Apyrase-treated WT, Gli-treated WT, *mrp4*, and *mrp5* cells compared to evk-treated WT control cells. Error bars denote ± SEM, *n* ≥ 30, and means labeled with different letters are significantly different at *p* < 0.05, Dunnett’s C (variance not neat).

**Figure 5 ijms-23-09002-f005:**
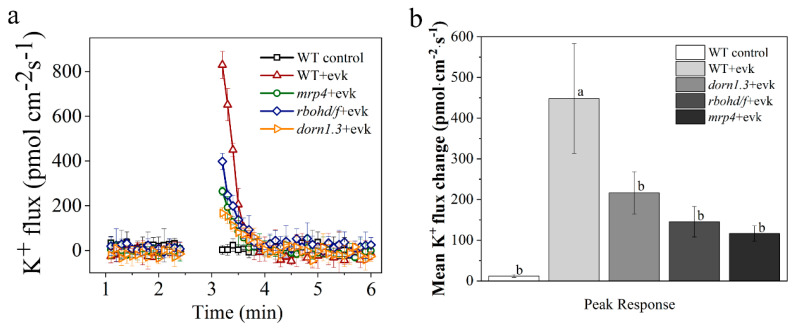
Evk-induced net K^+^ flux in WT, *mrp4*, *dorn1.3*, and *rbohf* mutant guard cells. (**a**) Net K^+^ flux in WT, *mrp4*, *dorn1.3*, and *rbohd/f* mutant guard cells. K^+^ flux was measured in guard cells for 3–5 min. Then, 8 μM evk was added to the test buffer. Each point was the mean of at least 3 individual guard cells and bars represent the standard error of the mean. (**b**) Mean K^+^ fluxes in WT, *mrp4*, *dorn1.3,* and *rbohd/f* mutant guard cells during the peak response periods. Error bars denote ± SEM, *n* ≥ 3, and means labeled with different letters are significantly different at *p* < 0.05, Dunnett’s C (variance not neat).

**Figure 6 ijms-23-09002-f006:**
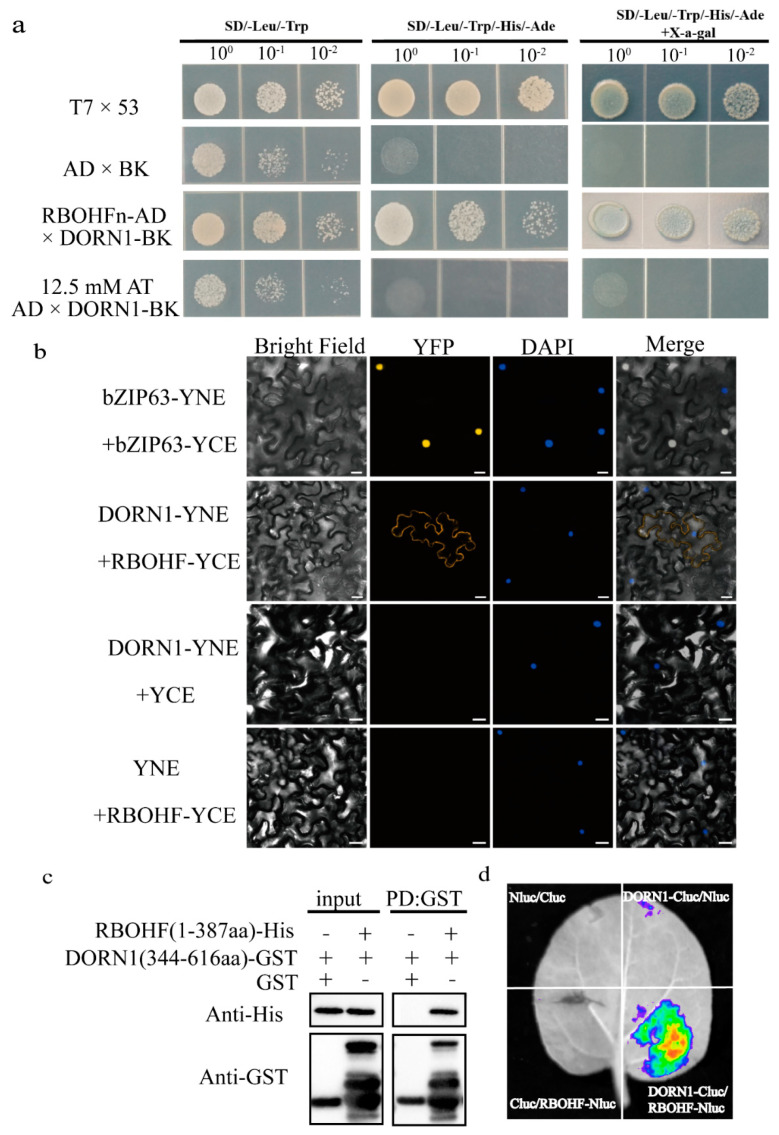
RBOHF interacts with DORN1. (**a**–**d**) Y2H, BiFC assay, LCI assays, and in vitro pulldown, respectively, show that RBOHF interacts with DORN1. (**a**) In the Y2H assays, the DORN1 was used as the bait vector. DORN1 has self–activation and can be inhibited by 12.5 mM 3-AT. Yeast clones were grown on SD/–Leu/–Trp and SD/–Ade/–His/–Leu/–Trp and SD/–Ade/–His/–Leu/–Trp with X-α-gal plate. When the self-activation is inhibited, the colony grows well on these plates. (**b**) In BiFC assay confirms the interactions between RBOHF and DORN1 on the plasma membrane. DAPI (blue) was applied to mark the nucleus. Bar = 25 μm. (**c**) GST–DORN1 can pull down the RBOHF with His tag, while GST-4T-1 cannot pull down the RBOHF. (**d**) In the LCI assay, only DORN1-Cluc/RBOHF-Nluc zone has stronger fluorescence than the other three control parts: Cluc/Nluc, DORN1-Cluc/Nluc, Cluc/RBOHF-Nluc.

**Figure 7 ijms-23-09002-f007:**
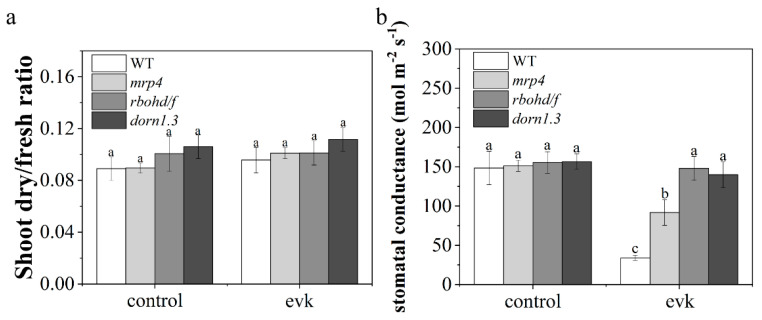
(**a**) The ratio of fresh weight to dry weight. In the WT control group, Arabidopsis plants were treated with ethyl alcohol fumigation, and in the WT evk group, Arabidopsis plants were treated with evk fumigation at 8 μM final concentration. Error bars denote ± SEM (standard error of mean), *n* = 6–8, *p* < 0.05, means labeled with different letters are significantly different at *p* < 0.05, Dunnett’s C (variance not neat). (**b**) Stomatal conductance in WT, mrp4, rbohd/f, and dorn1.3 plants before and after evk treatment for 1 h. The stomatal conductance was monitored with a diffusion porometer. The vertical scale represents the stomatal conductance before and after evk treatment. Data were obtained from at least 20 leaves from different Arabidopsis plants; bars represent the standard error of the mean.

**Figure 8 ijms-23-09002-f008:**
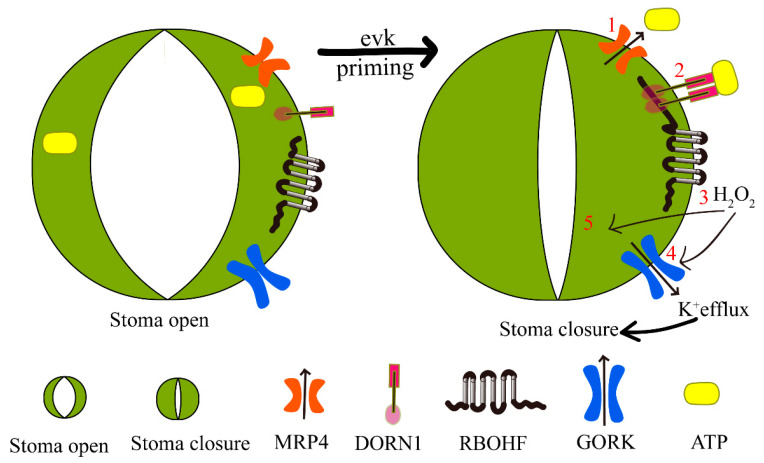
The proposed model depicting the mechanism of evk-induced stoma closure. The green part is kidney-shaped cells, which are surrounded by two kidney-shaped cells to form the organ stomata. Evk, as a signal substance, plays an early warning role, thus causing stomata to close. (**1**). Evk-induced eATP accumulation by MRP4; (**2**). eATP receptor DORN1 accepted eATP; (**3**). DORN1 regulate RBOHF to burst H_2_O_2_; (**4**). H_2_O_2_ regulates K^+^ efflux to cause stoma closure (**5**).

**Table 1 ijms-23-09002-t001:** Primer sequence.

Gene Name	Forward Primer Sequence (5′–3′)	Reverse Primer Sequence (5′–3′)
*DORN1*	AACCACTCACCTTACGCTTGG	AGTCGCCGTTTTTCCCTCT
*APY1*	ACACGATGAAAAACCACGAGG	AAGAGTTTGCTGATTGCCGAG)
*APY2*	GGATAACCATCAACGCACTAAAAG	GGGACGAACTGTAGCAACAGG
*MRP4*	CAAATCTCCACTGACGCTCG	CCAACGTACATGACGCCGA
*MRP5*	GCCGCAGTTACATTCGCTAC	CCAGATCAGGAAAGTTCCGAAG
*RBOHF*	TTCGCATCATTTGTTCGTCA	TGTAGCGTTAGAACATTACCAGGA
*RBOHD*	ATCAAGGTGGCTGTTTACCC	GGGAGCTGATGTGATTGAGA
*ACTIN2*	AGTGGTCGTACAACCGGTATTGT	GATGGCATGAGGAAGAGAGAAAC
*EF1α*	TCCAGCTAAGGGTGCC	GGTGGGTACTCGGAGA

## Data Availability

The data that support the findings of this study are available from the corresponding author upon request.

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
