# Peer review of "Ethyl Vinyl Ketone Activates K+ Efflux to Regulate Stomatal Closure by MRP4-Dependent eATP Accumulation Working Upstream of H2O2 Burst in Arabidopsis"

_ijms, 2022, doi:10.3390/ijms23169002_

Round 1

Reviewer 1 Report

It is interesting new results, however, significant corrections and furter explanations are required to make text more clear.

Abstract:

It should be understandable for the readers who do not know all the details, but the current version contains many abbreviations that are clear only to „deep-expert“ in the field. Please, re-write to avoid abbreviations and make the text understandable.

Line 50: Hand must be with capital, it is the name!

Line 50: do you mean soybean homogenate, not „homogeneous soybean?

Line 51: „When soybeans were dependent on 13-HOPT and LOX activity was blocked, they were able to synthesize evk“ – please, re-formulate.

Line 283: „RBOHF interacts with DORN1“ – this is only indirect evidence. It will be nice to show first co-localization of both proteins in the same domain.

Lines 315-319: please, re-write this part more logicaly.

Lines 397 – 401: very confusing. How you can add 8 µM evk to soil? You can not compare plants on agar (which agar, which conditions?) and in soil.

How did you culture Nicotiana?

Line 404: „ starting plants“ ?? You can tell baout plants because you used isolated leaves.

Line 405: „were fixed“ – what do you mean as fixed?

Lines 413-425: text is unclear. How 10 days old seedlings can be soaked in 100 µl of the solution? How you can collect samples after 1 minute of incubation from the large number of the treatments?

  Line 428: „Total RNA was isolated“ ? From stomata cell or from whole seedlings?

Lines 435-447: unclear descriptions: it seems you used multiply treatments and finally dye fluoresence did not reflecte real H2O2 in planta. Please, have a look here:

dx.doi.org/10.17504/protocols.io.bx49pqz6

Lines 448-449: the concentrations have been used are non-specific. Please, explain why you choosed it.

Figure 4, panels c, d: please, explain what do you mean as % of increase ?

Author Response

Manuscript ID: ijms-1803124

Title: Ethyl vinyl ketone activates K+ efflux to regulate stomatal closure by MRP4-dependent eATP accumulation working upstream of H2O2 burst in Arabidopsis

Dear Reviewer,

I would like to take this great opportunity to thank you for your precious comments and advice. Those comments are all valuable and very helpful for revising and improving our paper. We have studied the comments carefully and have made corrections which we hope meet with approval. I believe that the addressing of these comments has greatly improved the quality of this manuscript.

Question 1: It should be understandable for the readers who do not know all the details, but the current version contains many abbreviations that are clear only to “deep-expert” in the field. Please, re-write to avoid abbreviations and make the text understandable.

Answer: Thank you for your valuable comments. I have put the abbreviation and the corresponding full name in the original text (Line 30-31).

Question 2: Line 50: Hand must be with capital, it is the name!

Answer: Thank you for your suggestion. I have revised it (Line 51)

Question 3: Line 50: do you mean soybean homogenate, not „homogeneous soybean?

Answer: Yes, it is soybean homogenate. Thank you for your suggestion. I have revised it (Line 51).

Question 4: Line 51: When soybeans were dependent on 13-HOPT and LOX activity was blocked, they were able to synthesize evk–please, re-formulate.

Answer: Thank you for your advice. I have revised it. When LOX activity was blocked, soybeans were able to synthesize evk in a 13-HOPT-dependent way (Line 51-53).

Question 5: Line 283: “RBOHF interacts with DORN1” – this is only indirect evidence. It will be nice to show first co-localization of both proteins in the same domain.

Answer: Both proteins are located on the plasma membrane [1, 2, 3]. Therefore, there is no repetition of protein localization in this paper. In our paper, through BiFC experiment, connect the two proteins to the C/N ends of YCF respectively (DORN1-YNE and RBOHF-YCE). Once two proteins interact with each other and then reduce the distance in space, then YCF will be detected with yellow fluorescence, and the experiment shows that this yellow fluorescence is located in the plasma membrane. It further shows that these two proteins can interact on the plasma membrane.

Question 6: Lines 315-319: please, re-write this part more logicaly.

Answer:  Thank you for your suggestion. I have revised the original text (Lines 337-343)

Question 7: Lines 397 – 401: very confusing. How you can add 8 µM evk to soil? You can not compare plants on agar (which agar, which conditions?) and in soil.

Answer: Sorry, I didn't write in detail, which led to your misunderstanding. I have corrected it.

Firstly, I did not add 8 µM evk to soil. Detection of qRT-PCR, stomatal aperture measurements, K+ efflux was performed on 18-day-old Arabidopsis WT or mutant plants, which were planted in soil.  The final concentration of evk used in all experiments was 8 µM (Lines 434-438).

Secondly, detection of eATP content,reactive oxygen species (ROS) in guard cells was analyzed on 12-day-old Arabidopsis WT or mutant plants, which were grown on agar medium (1/2MS: 2% sucrose, 1.2% agar, pH 5.8). Plants were cultivated at 21–23°C, 70% relative humidity, and 80-110 mol m–2 s–1 light intensity under long-day (16 h light/8 h dark) conditions.

Question 8: How did you culture Nicotiana?

Answer: Sorry for not describing the planting of Nicotiana in detail, I have modified it (Lines 439-452).

Question 9: Line 404: “starting plants” ? You can tell about plants because you used isolated leaves.

Answer: Improper use of words leads to dyslexia. Thank you for your guidance. “starting plants” means: before the measurement of stomatal aperture, the stomata of plants were unified and completely open. I have deleted the word “starting” (Line 454-455).

Question 10: Line 405: “were fixed” – what do you mean as fixed?

Answer: Sorry, I didn't describe it clearly. Thank you for your advice. I have modified it. “fixed” means: to tear off the lower epidermis and lay it on the bottom of the dish. Fix the lower epidermis of the blade with adhesive tape (Line 455-456).

Question 11: Lines 413-425: text is unclear. How 10 days old seedlings can be soaked in 100 µl of the solution? How you can collect samples after 1 minute of incubation from the large number of the treatments?

Answer: Sorry, I didn't describe it clearly. Thank you for your advice. I have modified it.

Firstly, the seeds planted in the 96-well plate (one seedling per well) germinate in 3 days, and then grow for 7 days. Seedlings that grow for 10 days are actually only 7 days old. The seedlings are in a relatively small state, so the whole plant can be covered with 100ul of test solution (Line 465-467).

Secondly, collect the test solution after 1 minute soaking treatment, and there are a total of 6 repetitions. If you add liquid incubation at one time and collect it together, there will be errors. Therefore, each seedling is soaked separately, and the treatment of the next seedling and the collection of incubation liquid are not started until the end of collection.

Question 12: Line 428: “Total RNA was isolated” ? From stomata cell or from whole seedlings?

Answer: Sorry, I didn't describe it clearly. I have modified it. The extraction of total RNA here is from the leaves of Arabidopsis thaliana (Line 481).

Question 13: Lines 435-447: unclear descriptions: it seems you used multiply treatments and finally dye fluoresence did not reflecte real H2O2 in planta. Please, have a look here:

dx.doi.org/10.17504/protocols.io.bx49pqz6

Answer: Thank you for your valuable comments on the method description. I have described the determination method in detail. The samples on the slide will cause the fluorescence intensity to rise under the continuous laser irradiation, so the photos are collected on the machine after all of the processing (the H2DCFDA staining, inhibitor pretreatment and evk treatment), the sample will be placed on the microscope and observe samples under an microscope with excitation at 490 nm and emission at 530 nm, which will not keep the sample under excitation light all the time (Line 494-496).

Question 14: Lines 448-449: the concentrations have been used are non-specific. Please, explain why you choosed it.

Answer: I'm sorry that I didn't attach the cited literature. I've modified it and attached the corresponding literature. These concentrations are based on published papers. The concentrations of DPI, PPADS, Apyrase and Gli in the experiments were 50 µM [4], 100 µM [5], 25 µM [6] and 1.5 µM [7], respectively.

Question 15: Figure 4, panels c, d: please, explain what do you mean as % of increase ?

Answer: Thank you for your reminder, (fluorescence value of treatment group - fluorescence value of control group) / fluorescence value of control group=% of increase

Reference:

[1] Bouwmeester, K.; Han, M.; Blanco-Portales, R.; Song, W.; Weide, R.; Guo, L.Y.; van der Vossen, E.A.G.; Govers, F. The Arabidopsis lectin receptor kinase LecRK-I.9 enhances resistance to Phytophthora infestans in Solanaceous plants. Plant biotechnology journal, 2014, 12(1):10-16.

[2] Bouwmeester, K.; de Sain, M.; Weide, R.; Gouget, A.; Klamer, S.; Canut, H.; Govers, F. The lectin receptor kinase LecRK-I.9 is a novel Phytophthora resistance component and a potential host target for a RXLR effector. PLoS pathogens, 2011, 7(3), e1001327.

[3] Di, Q.; Li, Y.; Li, S.; Shi, A.; Zhou, M.; Ren, H.; Yan, Y.; He, C.; Wang, J.; Sun, M.; Yu, X. Photosynthesis Mediated by RBOH-Dependent Signaling Is Essential for Cold Stress Memory. Antioxidants, 2022, 11(5).

[4] Chun, Y.J.; Gong, J.Q.; Guo, Z.J.; Li, S.W.; Zuo, Y.X.; Shen, Y.B. Linalool Activates Oxidative and Calcium Burst and CAM3-ACA8 Participates in Calcium Recovery in Arabidopsis Leaves. International Journal of Molecular Sciences, 2022, 54, 23(10):5357.

[5] Clark, G.; Fraley, D.; Steinebrunner, I.; Cervantes, A.; Onyirimba, J.; Liu, A.; Torres, J.; Tang, W.; Kim, J.; Roux, S.J. Extracellular nucleotides and apyrases regulate stomatal aperture in Arabidopsis. Plant physiology, 2011, 156(4):1740-1753.

[6] Hao, L.H.; Wang, W.X.; Chen, C.; Wang, Y.F.; Liu, T.; Li, X.; Shang, Z.L. Extracellular ATP Promotes Stomatal Opening of Arabidopsis thaliana through Heterotrimeric G Protein α Subunit and Reactive Oxygen Species. Molecular Plant, 2012, 5(4):852-864.

[7] Martinoia, E.; Klein, M.; Geisler, M.; Bovet, L.; Forestier, C.; Kolukisaoglu, Ü.; Müller-Röber, B.; Schulz, B. Multifunctionality of plant ABC transporters – more than just detoxifiers. Planta, 2001, 214(3):345-355.

Reviewer 2 Report

Manuscript "Ethyl vinyl ketone activates K+ efflux to regulate stomatal closure by MRP4-dependent eATP accumulation working up

The stream of H2O2 burst in Arabidopsis by Junqing Gong, Lijuan Yao, Chunyang Jiao, Zhujuan Guo , Shuwen Li , Yixin Zuo and Yingbai Shen describes the details of hydrogen peroxide localization in stomatal guard cells. The work was done carefully, contains a sufficient number of controls and a qualitative discussion. However, there are several issues that remain unclear. Normally, stomatal regulation is associated with transpiration and respiration. These processes were ignored in this work. Binding to potassium used in the article is acceptable, but still somewhat remotely related to the described processes. Perhaps the authors should highlight the role of respiration in the discussion of this study.

In the introduction, the research methodology is considered quite broadly. However, a clearly formulated task: What kind of hypothesis was tested by the authors cannot be found in the formulated form. It is better to transfer this part to the discussion and formulate it again.

Figures 1, 2 and 3 (histograms) are best done in doubles.

Figure 5a is practically unreadable, it should be made in color and enlarged, since it is not possible to make out something there.

The article lacks information on confirming changes in water evaporation (well, at least dry/wet weight of leaves) and respiration, so the conclusions and the final scheme look far-fetched, which needs to be explained.

The final scheme also raises questions, which can mislead the reader. Firstly, it is not clear why two cells are combined into one, which is not true, and secondly, it is not clear which potassium channels are in question in the figure. The image gives the impression that we are talking about plasmodesmata. Other components, by which the authors apparently meant membrane proteins, are drawn so that they can also be confused with cellular compartments, for example, with ER. It is also not clear why the same symbol is used for the loss of water and for the penetration of bacteria, because the water comes out and the leaf. I think that it is worth changing the figure to a fragment of the scheme of action and clarifying the structure of processes with a membrane and a lipid bilayer, so that it is clear what is meant.

I think that the conclusion section is a fragment of the discussion.

In conclusion, it is necessary to try to formulate how to find out the pattern more deeply, taking into account those cells that actually provide the inflow and outflow of potassium and provide this process with energy, which is associated with the transport of sugars and respiration.

The article is certainly interesting and contains fresh material. That is why it requires special care in its design and evidence. Hopefully this will be easily fixed.

Author Response

Manuscript ID: ijms-1803124

Title: Ethyl vinyl ketone activates K+ efflux to regulate stomatal closure by MRP4-dependent eATP accumulation working upstream of H2O2 burst in Arabidopsis

Dear Reviewer,

I would like to take this great opportunity to thank you for your precious comments and advice. Those comments are all valuable and very helpful for revising and improving our paper. We have studied the comments carefully and have made corrections which we hope meet with approval. I believe that the addressing of these comments has greatly improved the quality of this manuscript.

Question 1: In the introduction, the research methodology is considered quite broadly. However, a clearly formulated task: What kind of hypothesis was tested by the authors cannot be found in the formulated form. It is better to transfer this part to the discussion and formulate it again.

Answer: Thank you for your valuable suggestions. We apologize for the confusion generated by the previous version of the discussion and sincerely hope that our logic is now easier to follow with this new version (Line 337-410).

Question 2: Figures 1, 2 and 3 (histograms) are best done in doubles.

Answer: Thank you for your positive comments and suggestions. I have revised them all and put them back in the corresponding position of the text (Line 123-124; Line 142-143; Line 188-189).

Question 3: Figure 5a is practically unreadable, it should be made in color and enlarged, since it is not possible to make out something there.

Answer: We agree with this suggestion and have modified figures (5a). The picture has been colored and enlarged. So that pictures can convey clearer information (Line 278-279).

Question 4: The article lacks information on confirming changes in water evaporation (well, at least dry/wet weight of leaves) and respiration, so the conclusions and the final scheme look far-fetched, which needs to be explained.

Answer: Thank you for pointing this out. We agree that more study about dry/wet weight of leaves and respiration would be useful to clarify the mechanism. We have supplemented this part of data. The data of dry and fresh weight and stomatal conductance are as follows, and have been put into the original text (Line 318-335).

Firstly, after fumigation with evk, there was no significant difference in dry/fresh ratio between evk and control group, indicating that evk did not cause stress on plant growth. Secondly, compared with the control group, the stomatal conductance of WT evk treatment group decreased significantly, the stomatal conductance of the mutant also decreased, but the degree of decline was less than that of the wild type. Measurements of stomatal conductance showed similar results to the stomatal aperture experiments (Fig.1a). Evk not only does not affect plant growth, but also mediates stomatal closure. It also shows that evk is a qualified stomatal switch regulator.

Question 5: The final scheme also raises questions, which can mislead the reader. Firstly, it is not clear why two cells are combined into one, which is not true, and secondly, it is not clear which potassium channels are in question in the figure. The image gives the impression that we are talking about plasmodesmata. Other components, by which the authors apparently meant membrane proteins, are drawn so that they can also be confused with cellular compartments, for example, with ER. It is also not clear why the same symbol is used for the loss of water and for the penetration of bacteria, because the water comes out and the leaf. I think that it is worth changing the figure to a fragment of the scheme of action and clarifying the structure of processes with a membrane and a lipid bilayer, so that it is clear what is meant.

Answer: Thank you for your valuable comments on my model diagram. There are some misunderstandings in the pattern diagram. Thank you for your detailed guidance and suggestions. I will correct or explain them one by one (Line 416-422).

Firstly, the green part is kidney shaped cells, which are surrounded by two kidney shaped cells to form the organ stomata. It's not two cells merging into one cell. Evk, as a signal substance, plays an early warning role, thus causing stomata to close.

Secondly, proteins (MRP4, DORN1, RBOHF, GORK) are located in the plasma membrane and have nothing to do with the endoplasmic reticulum (ER).

Thirdly, K+ efflux plays a key role in pore closure. The foothold of this paper is that evk causes eATP accumulation, which stimulates the burst of H2O2, and finally leads to K+ efflux, which leads to stomatal closure. This is a complete biological process, which explains why evk, a signal substance, can induce the mechanism of stomatal closure. We guess that GORK, an outward K+ channel, plays a very important role here. In the pattern diagram, I have marked GORK in it.

Finally, the biological significance of stomatal closure is to reduce water loss and pathogen invasion. In order to avoid misleading readers, I deleted the symbols of water loss and pathogens.

Question 6: I think that the conclusion section is a fragment of the discussion.

Answer: Thank you. I have merged this part into the conclusion (Line 411-415).

Question 7: In conclusion, it is necessary to try to formulate how to find out the pattern more deeply, taking into account those cells that actually provide the inflow and outflow of potassium and provide this process with energy, which is associated with the transport of sugars and respiration.

Answer: Thank you for your valuable advice. Due to the pressure of graduation and the urgency of time, I will take this part as the next research direction, such as how to provide the energy source of potassium ion inflow and outflow, and how to prove that it is related to sugar and respiration.

Round 2

Reviewer 1 Report

Thank you!

The text became better, but require more corrections/explanations.

Here is some points:

Line 135: inhibiter?

In your case DPI can not be consider as inhibitor, it is inhibitor of plant NADPH osidase at 0.5 mkM, but lethal for plants at 2 mkM. In some cases even pico molar concentration is specific.  

https://www.ncbi.nlm.nih.gov/pmc/articles/PMC7295061/

You can not just cited somebody and simple copy-paster, it is really necesssary to make dose-response curve to get specific dose.

Line 212: biological or technical? „had three replicates“.

Lines 223 and 238: the same sentence: (9.04% above control).

This title was wrong: you did not study effect of H2O2: „H2O2 effects on eATP secretion during evk-induced stomatal closure“.

Line 385: „Arabidopsis plants were treated“ – how? Sprayed?

Line 734: „50 mM H2DCF-DA“ – or mkM? This is too high, normal concentration should be 2 mkM.

Lines 730-743: dark or light? Where is dye loading and de-esterfication control? Without such control you can not measure..

Lines 687- 689: „All seedings planted in the 96-well plate (one seedling per well) germinate in 3 days, 687 and then grow for 7 days. Seedlings that grow for 10 days are actually only 7 days old. 688 The seedlings are in a relatively small state, so the whole plant can be covered with 100uL 689 of test solution“.  – 7 days old seedlings have a root length at least 3 cm and corresponding shoots. How can this can be covered by 100 mkl? What is the ratio between medium and plants?  Why did you use toxic solution for plant growth?

Reviewer 2 Report

It is gratifying to see that the authors took the comments expressed critically and reworked the manuscript quite deeply and eliminated all inaccuracies. I believe that the manuscript can be recommended for publication in a International Journal of Molecular Sciences in its present form.

Author Response

Suggestion: It is gratifying to see that the authors took the comments expressed critically and reworked the manuscript quite deeply and eliminated all inaccuracies. I believe that the manuscript can be recommended for publication in a International Journal of Molecular Sciences in its present form.

Reply: Thank you for your guidance and affirmation. I will conduct in-depth research on the two aspects of respiration and photosynthesis you proposed in future research. After all, stomata and photosynthetic respiration are closely related. Thank you again for your guidance.

Round 3

Reviewer 1 Report

Thank you very much for clarification. It is better now, but some points need further clarity.

Line 133: „inhibiter“ – must be inhibitor. Moreover, you can not simple „consult“ literature without testing specific concentration yourself. There are so many wrong points published…

For check specific concentration one need to do own dose-response curve and find concentration what have effect on NADPH oxidase activity. In the case of DPI this concentration is 0.1 – 1 micromol. Higher concentration have a lot of side effects and can be linked to effect of DPI, but not as effect of NADPH oxidase inhibition.

You have to mention this in discusion (at least).

For the ROS measure, please also „consult“ this protocol dx.doi.org/10.17504/protocols.io.bx49pqz6.

For the 96-well plates, plants looking very stressed with disturbing metabolism. So, any measurements and data interpretation should be consider with precaution. This should be mentioned in discussion.

Author Response

Thanks for the reviewers’ constructive comments concerning our article (Manuscript ID: ijms-1803124). I would like to take this great opportunity to thank you for your precious comments and advice. Those comments are all valuable and very helpful for revising and improving our paper, especially the emphasis on ROS determination and DPI use has benefited me a lot. We have studied the comments carefully and have made corrections which we hope meet with approval. I believe that the addressing of these comments has greatly improved the quality of this manuscript. In this revised manuscript, changes were marked up using the “Track Changes” function. Point-by-point responses to the reviewers are listed below this letter. I would like to show details as follows:

Question1: Line 133: „inhibiter“ – must be inhibitor. Moreover, you can not simple „consult“ literature without testing specific concentration yourself. There are so many wrong points published…

For check specific concentration one need to do own dose-response curve and find concentration what have effect on NADPH oxidase activity. In the case of DPI this concentration is 0.1 – 1 micromol. Higher concentration have a lot of side effects and can be linked to effect of DPI, but not as effect of NADPH oxidase inhibition.

You have to mention this in discusion (at least).

Answer: Thank you very much for your questions about the use of DPI concentration. I agree with you very much. It is true that whether the use of DPI at high concentration will poison plants is not an inhibitor. In future research, I will pay close attention to this key issue you mentioned. Thank you for your guidance.

A previous study demonstrated that DPI at subpicomolar concentrations (10-13 to 10-14 M) exhibits specific effects against NOX2 in primary midbrain neuron-glia cultures [1]. The ultralow doses of DPI have high efficiency and low toxicity and may become a potential treatment method in clinic in the future [2]. But what is the dosage of DPI in plants? Through consulting the literature, we found that the concentration of NADPHase inhibition in plants is 100 µM [3], 10 µM [4], 50 µM [5]. After WT was pretreated with DPI at different concentrations, it was treated with 8 µM evk for 1h to see the stomatal aperture. It was found that the effect of evk on stomatal closure could not be inhibited at the DPI concentration of 50 µM(tupianyuanwen). Will the concentration in plants be too high? Will it lead to a toxic effect rather than a real inhibitory effect? This part needs further verification. (Line: 413-423 )

Question2: For the ROS measure, please also „consult“ this protocol dx.doi.org/10.17504/protocols.io.bx49pqz6.

Answer: As for the determination method of ROS, I will refer to the literature you mentioned in the future research. In this article, because of the pressure of graduation, I have no time to do this experiment again. I hope you will understand. Thank you for your careful guidance. I have learned a lot and will have great guiding significance for me in future research.

For the determination of ROS, it is necessary to complete the H2DCF-DA concentration specific reaction curve, as well as the dyeing process and the esterification process. (Line: 423-425 )

Question3:  For the 96-well plates, plants looking very stressed with disturbing metabolism. So, any measurements and data interpretation should be consider with precaution. This should be mentioned in discussion.

Answer: Determine the effect of evk on plant eATP concentration. Planting plants in 96 well plates will have an impact on plant growth, which is not conducive to plant growth. The nutrients obtained by the roots are very limited. Due to the space problem, only 180 ul of culture medium is added, so it will be subject to nutrient stress, and the overall growth of plants is relatively small. The advantage of this method is that it can detect the release of eATP concentration of seedlings by detecting the eATP in the living state, but its limitations cannot be ignored, and it needs to be improved here. Since we have made the control group, that is, the seedlings planted in the 96 well plate, but the only difference between the control group and the experimental group is whether evk is added, so the main data obtained is the research on evk effect. However, the problem of the growth restriction of plants from the narrow space of the 96 well plate cannot be ignored, and I need to propose better research methods. (Line: 426-437 )

[1] Kumar, A.; Barrett, J.P.; Alvarez-Croda, D.M.; Stoica, B.A.; Faden, A.I.; Loane, D.J. NOX2 drives M1-like microglial/macrophage activation and neurodegeneration following experimental traumatic brain injury. Brain Behavior & Immunity, 2016, 291-309.

[2] Kuai, Y.; Liu, H.; Liu, D.; Liu, Y.; Sun, Y.; Xie, J.; Sun, J.; Fang, Y.; Pan, H.; Han, W. An ultralow dose of the NADPH oxidase inhibitor diphenyleneiodonium (DPI) is an economical and effective therapeutic agent for the treatment of colitis-associated colorectal cancer. Theranostics, 2020, 10(15):6743-6757.

[3] Melillo, M.T.; Leonetti, P.; Bongiovanni, M.; Castagnone-Sereno, P.; Bleve-Zacheo, T. Modulation of reactive oxygen species activities and H2O2 accumulation during compatible and incompatible tomato–root-knot nematode interactions. New Phytologist, 2006, 170(3):501-511.

[4] Li, J.; Jia, H.; Wang, J.; Cao, Q.; Wen, Z. Hydrogen sulfide is involved in maintaining ion homeostasis via regulating plasma membrane Na+/H+ antiporter system in the hydrogen peroxide-dependent manner in salt-stress Arabidopsis thaliana root. Protoplasma, 2014, 251(4):899-912.

[5] Wen, F.; Xing, D.; Zhang, L. Hydrogen peroxide is involved in high blue light-induced chloroplast avoidance movements in Arabidopsis. Journal of Experimental Botany, 2008, (10):2891-2901.

Round 4

Reviewer 1 Report

Thank you! Please, correct line 133 (inhibitor, not inhibiter).

For the future, you can "consult" only relevant literature. One what is not determine IC50 for specifci enzyme (like in the case of DPI) can not be relevant and must be ignored. Also, control mean optimal conditions, not stressed one. 

Very imortant information seems to be mising: which medium you have used for plants (line 592)? Did you add sucrose? How many macroinons have been used?  For the small volume proper medium should be used, not toxic one, with 6 macro ions only. It is not acceptable to used medium for callus (like MS) for the plants in uch a small volume. You induced a very rapid P deficeincy and Cl toxicity...

Please, provide this information. The rest part is Ok for this story, but missing information shuld be added.

My best regards!
